# Variant Landscape of 15 Genes Involved in Corneal Dystrophies: Report of 30 Families and Comprehensive Analysis of the Literature

**DOI:** 10.3390/ijms24055012

**Published:** 2023-03-06

**Authors:** Di Zhu, Junwen Wang, Yingwei Wang, Yi Jiang, Shiqiang Li, Xueshan Xiao, Panfeng Wang, Qingjiong Zhang

**Affiliations:** State Key Laboratory of Ophthalmology, Zhongshan Ophthalmic Center, Sun Yat-sen University, Guangdong Provincial Key Laboratory of Ophthalmology and Visual Science, Guangzhou 510060, China

**Keywords:** corneal dystrophies, variants spectrum, *TGFBI*, *CHST6*, *SLC4A11*, *ZEB1*

## Abstract

Corneal dystrophies (CDs) represent a group of inherited diseases characterized by the progressive deposit of abnormal materials in the cornea. This study aimed to describe the variant landscape of 15 genes responsible for CDs based on a cohort of Chinese families and a comparative analysis of literature reports. Families with CDs were recruited from our eye clinic. Their genomic DNA was analyzed using exome sequencing. The detected variants were filtered using multi-step bioinformatics and confirmed using Sanger sequencing. Previously reported variants in the literature were summarized and evaluated based on the gnomAD database and in-house exome data. In 30 of 37 families with CDs, 17 pathogenic or likely pathogenic variants were detected in 4 of the 15 genes, including *TGFBI*, *CHST6*, *SLC4A11*, and *ZEB1*. A comparative analysis of large datasets revealed that 12 of the 586 reported variants are unlikely causative of CDs in monogenic mode, accounting for 61 of 2933 families in the literature. Of the 15 genes, the gene most frequently implicated in CDs was *TGFBI* (1823/2902, 62.82% of families), followed by *CHST6* (483/2902, 16.64%) and *SLC4A11* (201/2902, 6.93%). This study presents, for the first time, the landscape of pathogenic and likely pathogenic variants in the 15 genes responsible for CDs. Awareness of frequently misinterpreted variants, such as c.1501C>A, p.(Pro501Thr) in *TGFBI*, is crucial in the era of genomic medicine.

## 1. Introduction

Corneal dystrophies (CDs) are a group of rare hereditary disorders characterized by an abnormal deposition of corneal material in different layers of the cornea. The prevalence of CDs is about 1‰ (897 per million) [1]. Based on the anatomical location of the corneal deposition, clinical and pathological phenotypes, and genetic defects, it has been classified into four main categories with 22 subtypes by the International Committee for Classification of Corneal Dystrophies (IC3D) [2,3]. The gradual accumulation of deposits in different corneal layers leads to progressive visual loss and even blindness. CDs are highly heterogeneous in terms of causative genes and patterns of inheritance, in which the autosomal dominant trait is the most common form. To date, variants in at least 15 genes have been reported to cause Mendelian inheritance of CDs, including *TGFBI* [4], *CHST6* [5], *SLC4A11* [6], *TACSTD2* [7], *ZEB1* [8], *UBIAD1* [9], *KRT12* [10], *COL8A2* [11], *OVOL2* [12], *PIKFYVE* [13], *COL17A1* [14], *GRHL2* [15], *AGBL1* [16], *DCN* [17], and *KRT3* [10]. The specific genotype–phenotype correlation has been widely reported [2,3,18,19,20]. Due to a lack of effective conservative therapy, a corneal transplant is often considered the ultimate treatment. However, such treatment for patients with CDs is frequently challenged by a shortage of donor corneas and a high graft failure rate [21,22]. In the era of next-generation sequencing [23,24], identifying pathogenic variants in genes contributing to CDs helps provide confident genetic counseling, an alternative approach to managing diseases that family members at risk may benefit from [25]. Recognizing the characteristics of pathogenic and likely pathogenic variants in these genes would be valuable for clinical gene testing for CDs. Although many variants in the 15 genes have been reported in patients with CDs, there is a lack of a comprehensive summary of these variants, especially comparative analysis of large datasets. Cross-reference variant information with large databases (online and/or in-house) has shown that non-pathogenic variants might be mistakenly described as pathogenic [26,27,28]. Similarly, our previous studies have demonstrated that such analyses not only provide a landscape of pathogenic variants but also reveal misinterpreted variants in a subset of genes [29,30]. These kinds of problems have become more frequent, since the detection and reporting of variants are increasingly easy for many clinicians, even without the necessary knowledge of medical genetics. In the current study, variants in the 15 genes were analyzed based on 37 families with CDs and 10493 families with other eye conditions. A comparative analysis of variants in the 15 genes was performed using large datasets (our in-house data, variants recorded in HGMD, gnomAD, and some recently reported literature). According to our results, two factors played unique roles in elucidating misinterpreted variants, including known phenotypes in our in-house dataset and carrier frequency (CF) in large databases [28]. Data from our study not only demonstrate a landscape of pathogenic and likely pathogenic variants in the 15 genes responsible for CDs but also reveal a few benign variants mistakenly interpreted as pathogenic variants. The results are valuable for clinical gene testing, which is widely performed as a routine procedure nowadays.

## 2. Results

### 2.1. Genes and Pathogenic or Likely Pathogenic Variants of CDs in Our In-House Database

In total, 17 pathogenic or likely pathogenic variants in 4 of the 15 genes were detected in 30 of 37 families with CDs, including 6 variants in *TGFBI*, 7 in *CHST6*, 3 in *SLC4A11*, and 1 in *ZEB1* (Figure 1 and Table 1). Among these variants, the following were novel: c.1096G>T, p.(Glu366*) in *CHST6*; c.110C>A, p.(Ser37*) and c.1766dupA, p.(Tyr589*) in *SLC4A11*; and c.1481_1482insTTTT, p.(Lys494Asnfs*12) in *ZEB1*. Pathogenic or likely pathogenic variants in *TGFBI*, *CHST6*, *SLC4A11*, and *ZEB1* accounted for 70% (21/30 families), 20% (6), 6.67% (2), and 3.33% (1) of families, respectively (Figure 2A). Most variants in *TGFBI* were missense variants, while most variants in *CHST6*, *SLC4A11*, and *ZEB1* were splicing variants, frameshift variants, and nonsense variants (Figure 2B). These variants were confirmed using Sanger sequencing and pedigree segregation analysis conducted in probands and their available family members (Figure 1 and Appendix A).

Detailed clinical information was available for 32 patients from 29 of the 30 families (Table 2); the patient (F29-V:1) from the remaining family was diagnosed with CD but her detailed clinical information was unavailable. Age at exam ranged from 6 months to 66 years old, with a median age of 31 years old for 32 patients with variants in the four genes, while it ranged from 1.5 years old to 66 years old with a median age of 32 years old for 23 patients with variants in *TGFBI* (Figure 2C). Visual acuity ranged from counting fingers to 1.0, and it declined gradually with age (Figure 2D). The first symptom was poor vision in 68.75% (22/32) of patients, ocular pain in 18.75% (6/32) of patients, and cornea opacities in 6.25% (2/32) of patients. Patients who complained of sore eyes were found to co-exist with punctate corneal epithelial defects and even corneal ulcers. All patients showed corneal opacity under slit-lamp, which contributed to their diagnosis (Figure 3). Corneal changes in patients demonstrated typical CDs. The patients with *TGFBI* c.370C>T, p.(Arg124Cys) presented the LCD (lattice corneal dystrophy) phenotype; the patients with *TGFBI* c.371G>A, p.(Arg124His) had GCD2 (granular corneal dystrophy, type 2); and the patients with *TGFBI* c.1663C>T, p.(Arg555Trp) showed the phenotype of GCD1 (granular corneal dystrophy, type 1). Furthermore, among eight probands with the c.371G>A, p.(Arg124His) variant in *TGFBI*, four (4/8, 50%) had both granular corneal dystrophy and high myopia, and two (2/8, 25%) had both granular corneal dystrophy and glaucoma. One proband with the c.1481_1482insTTTT, p.(Lys494Asnfs*12) variant in *ZEB1* had both cystic changes of the posterior corneal lamina and high myopia.

### 2.2. Genes and Variants of CDs in the Published Literature

In total, 586 variants in the 15 genes of CDs were reported in 2933 families with CDs (Appendix A). Variants in *TGFBI*, *CHST6*, *SLC4A11*, *TACSTD2*, *ZEB1*, *UBIAD1*, *KRT12*, *COL8A2*, *OVOL2*, *PIKFYVE*, *COL17A1*, *GRHL2*, *AGBL1*, *DCN*, and *KRT3* accounted for 63.08% (1850/2933), 16.26% (477), 6.78% (199), 4.36% (128), 2.59% (76), 2.52% (74), 1.06% (31), 0.95% (28), 0.78% (23), 0.41% (12), 0.41% (12), 0.20% (6), 0.27% (8), 0.17% (5), and 0.14% (4) of families, respectively. Of these, variants in most of these genes contributed to autosomal dominant CDs, while those in *CHST6*, *SLC4A11*, and *TACSTD2* were associated with autosomal recessive CDs. Variants in some of these genes were usually reported in Caucasian patients, while those in others were commonly detected in Asian patients. All families with variants in *OVOL2* or *GRHL2* and most of the reported families with variants in *ZEB1* (83.12%), *UBIAD1* (70.27%), *PIKFYVE* (91.67%), *COL17A1* (83.33%), *AGBL1* (75%), and *KRT3* (75%) were Caucasian, while most of the reported families with variants in *TGFBI* (76.86%), *CHST6* (72.88%), *SLC4A11* (85.97%), or *TACSTD2* (92.97%) were Asian. In addition, a small number of African patients were reported to have variants in *TGFBI* (0.37%), *CHST6* (1.66%), *SLC4A11* (1.49%), *TACSTD2* (3.13%), *UBIAD1* (1.35%), or *COL8A2* (7.14%) (Figure 4C). Furthermore, most reported variants were missense, but the main type of variants varied among these genes; missense variants in *TGFBI*, *CHST6*, and *SLC4A11* were the most common one, while splicing variants, frameshift variants, and nonsense variants were relative common in *TACSTD2* and *ZEB1* (Figure 4D).

All the reported variants in *CHST6*, *TACSTD2*, *UBIAD1*, *KRT12*, *PIKFYVE*, *GRHL2*, *AGBL1*, and *DCN* caused macular corneal dystrophy (MCD), gelatinous drop-like corneal dystrophy (GDLD), Schnyder corneal dystrophy (SCD), Messman corneal dystrophy (MECD), Fleck corneal dystrophy (FCD), posterior polymorphous corneal dystrophy (PPCD), Fuchs endothelial corneal dystrophy (FECD), and congenital stromal corneal dystrophy (CSCD), respectively. Variants in *SLC4A11* caused congenital hereditary endothelial dystrophy (CHED) or Harboyan syndrome (CHED co-existing with hearing impairment). A large proportion of variants in *ZEB1* caused PPCD, while a small proportion of variants in *ZEB1* caused FECD, and variant c.1920G>T, p.(Gln640His) caused CD co-existing with keratoconus [31,32]. Almost all the variants in *COL8A2* caused FECD, except for c.1363C>A, p.(Gln455Lys), which caused PPCD in a family [11]. All the variants in *OVOL2* caused PPCD, except for c.-339_361dup, which caused CHED [12]. Almost all the variants in *KRT3* caused MECD, except for a sporadic case with c.1492G>A, p.(Glu498Lys), which presented MECD co-existing with lattice corneal dystrophy (LCD) [33]. Almost all the variants in *COL17A1* caused epithelial recurrent erosion of dystrophy (ERED), except for a sporadic case with homozygous variant c.3554C>T, p.(Pro1185Leu), which presented FECD [34]. *TGFBI*, as a gene with a high degree of pleiotropy, causes various types of CDs, including epithelial basement membrane dystrophy (EBMD), Reis–Bucklers corneal dystrophy (RBCD), Thiel–Behnke corneal dystrophy (TBCD), LCD, granular corneal dystrophy type 1 (GCD1), and granular corneal dystrophy type 2 (GCD2). According to statistics, 49.46% (229/463) of reported LCD families carried the c.370C>T, p.(Arg124Cys) variant, while 87.61% (191/218) of reported GCD1 families carried the c.1663C>T, p.(Arg555Trp) variant, and 98.30% (753/766) of GCD2 families carried the c.371G>A, p.(Arg124His) variant.

### 2.3. Misinterpreted Variants in the Published Literature

The CF threshold was calculated based on the contributions of each gene according to our cohort study and literature summary (Figure 4A). Considering that about half of reported CDs were polygenic, the prevalence of monogenic CDs might be approximately around 450 (i.e., 897/2) per million. *TGFBI* accounted for no more than 63.15%, in which variants in three codons (i.e.,124, 555, and 626) accounted for the vast majority (86.86%, 1607/1850), while those in other codons only accounted for 13.14%. The sum of CF in other variants in *TGFBI* is expected to be less than 6/100,000. The CF thresholds for total pathogenic variants in *ZEB1*, *UBIAD1*, *KRT12*, *COL8A2*, *OVOL2*, *PIKFYVE*, *COL17A1*, *GRHL2*, *AGBL1*, *DCN*, and *KRT3* were estimated at around 1/100,000, 1/100,000, 0.5/100,000, 0.4/100,000, 0.4/100,000, 0.2/100,000, 0.2/100,000, 0.1/100,000, 0.1/100,000, 0.1/100,000, and 0.1/100,000, respectively. For *TGFBI* with a CF threshold of 6/100,000 outside of the three common ones, the c.337G>A, p.(Val113Ile); c.367G>C, p.(Asp123His); c.895G>A, p.(Asp299Asn); c.1501C>A, p.(Pro501Thr); c.1504A>G, p.(Met502Val); c.1631A>G, p.(Asn544Ser); and c.1998G>C, p.(Arg666Ser) variants were unlikely pathogenic due to their high allele frequency in gnomAD (38/280,598, 38/280,514, 131/279,704, 85/279,506, 494/279,598, 13/249,042, and 244/213,564, respectively). Moreover, the c.367G>C, p.(Asp123His) and c.1501C>A, p.(Pro501Thr) variants were detected in 51 and 83 families without the phenotype of CDs in our in-house database, respectively. The same condition was seen in *COL8A2* and *AGBL1* (Figure 5, Appendix A and Appendix A). In total, 12 variants from 61 families did not pass the CF threshold. They were likely benign or benign variants according to the ACMG/AMP classification (Appendix A) and were unlikely causative of CDs in monogenic mode.

Apart from those variants that may not be the cause of CDs, a total of 578 variants in these genes were identified in 2902 families, including those in our cohort and in the literature (Table 1, Figure 4B, and Appendix A). The percentage of three ethnic groups in these genes seemed not to change much (Figure 4C), and neither did the type of variants (Figure 4D).

## 3. Discussion

In the current study, 17 pathogenic or likely pathogenic variants (including 4 novel variants) in *TGFBI*, *CHST6*, *SLC4A11*, and *ZEB1* were detected in 30 of 37 families with CDs based on the systemic analysis of the exome sequencing data of the 15 genes related to CDs. Meanwhile, the variant landscape of the 15 genes in patients with CDs was characterized based on the comprehensive comparative analysis of our in-house data, HGMD data, gnomAD database, and literature reports. In addition, misinterpreted variants were also highlighted. Further evidence is expected to clarify the variants in question as well as genes with relatively rare reports. 

The detection of pathogenic or likely pathogenic variants in 4 of the 15 genes in 30 of 37 families with CDs suggests that most CDs (around 81.1%) can be resolved by analyzing these genes. The four genes with variants detected in our cohort were also ranked as the top five most implicated genes in CDs based on previous reports, indicating a common feature of genetic defects in different ethnic groups. The absence of variants in the other genes in our cohort may have partly been due to the small number of cases as well as the rarity of variants contributing to CDs. Misinterpreted variants and genes should also be taken into consideration. The landscape of pathogenic and likely pathogenic variants in the 15 genes involved in CDs was further characterized with the comprehensive analysis of 586 variants from 2933 families reported in the published literature and HGMD database and the comparative analysis of variants from large datasets (including all variants in the 15 genes from our in-house datasets of 10,530 individuals as well as those from the gnomAD database). Of these genes, *TGFBI*, *CHST6*, and *SLC4A11* are considered popular genes, and their distribution in the population decreased in turn. The distribution of each variant in different genetic groups is generally uniform [35]. However, each gene has its own genetic group distribution specificity, according to our combined data. *TGFBI*, *CHST6*, and *SLC4A11* were mainly found in Asian subjects, while *ZEB1*, *UBIAD1*, and *OVOL2* were mostly reported in Caucasian subjects. There were few reports on African populations. The top four genes mutated in CDs in Asian subjects were *TGFBI*, *CHST6*, *SLC4A11*, and *TACSTD2*, and the top four genes mutated for CDs in Caucasian subjects were *TGFBI*, *CHST6, TACSTD2*, and *ZEB1*. The ratio of pathogenic or likely pathogenic variants in *TGFBI*, *CHST6*, *SLC4A11*, and *ZEB1* in our cohort was consistent with the frequency in the population. On the other hand, such comparative analysis also revealed that misinterpreted variants in the genes involved in CDs are an important issue. Unawareness of this problem may lead to severe consequences in clinical practice in the era of genome medicine with widely applicable clinical gene tests. In the current study, at least 12 variants from 61 families were unlikely causative of CDs in monogenic mode (Appendix A), based on higher allele frequency in general population as well as unrelatedness with CDs in our cohort. Such kind of misinterpreted variants is expected to increase with time soon, since many unprofessional people are more likely to become involved in the detection, description, analysis, and report of not well-classified variants, given the increasing number of small commercial companies for exome sequencing, clinicians without necessary knowledge of medical genetics, journals lacking qualified review, etc. This kind of problem may be greatly reduced if ACMG/AMP standards and guidelines are well applied in the description of disease-associated variants. In addition, comparative analysis of large datasets can further elucidate misinterpreted variants, even variants with a series of previous reports. 

Variants in *TGFBI* are the most frequently reported monogenetic cause of CDs. TGFBI has highly conserved sequences across vertebrates [36]. The exact biochemical mechanisms causing various types of CDs have not been fully elucidated. Most of the reported *TGFBI*-related variants are enriched in exon 4 and exon 12 or FAS1-4 domains (Figure 5), and literature studies have reported and proposed this as a basis for measuring the pathogenicity of variants [37,38]. Although the reported variants c.367G>C, p.(Asp123His); c.1501C>A, p.(Pro501Thr); c.1504A>G, p.(Met502Val); and c.1631A>G, p.(Asn544Ser) are either located in the two exons or predicted to affect the FAS1-4 domain in *TGFBI* (Figure 5), they are unlikely pathogenic variants for the reasons mentioned above. Unlike many other genes, the classification of variants in *TGFBI* is challenging based on in silico online prediction tools, as these unlikely pathogenic variants mentioned above are hardly differentiate from the three common pathogenic variants in Arg124, Arg555, and His626 based on predicted damaging effects. Remembering this and seeking additional reliable evidence is crucial to reporting new pathogenic or likely pathogenic variants in future studies. To date, the pathogenicity of the three common pathogenic variants is largely based on studies of several large families, which is further supported by multiple CD families with these variants in the current study. Survivorship bias and the mutability of the underlying cytosine–guanine CpG in arginine codons were used to explain the high mutability of the Arg124 and Arg555 codons [39], and lysosomal biogenesis and autophagy were reported as the mechanism of (p.Gly623_His626del)-linked TBCD [40]. So far, there is no elucidated explanation for why popular areas of Arg124, Arg555, and His626 hot spots are highly mutable and pathogenic. Therefore, more research must be performed to reveal the mechanism of concentration of pathogenic variants in *TGFBI*.

## 4. Materials and Methods

### 4.1. Subjects 

Patients with different eye conditions were recruited from Pediatric and Genetic Clinic at the Zhongshan Ophthalmic Centre, Guangzhou, China. Informed consent was given by patients or their guardian as well as available family members before we collected their clinical data and peripheral venous blood samples. Genomic DNA was extracted from leukocytes of peripheral venous blood samples.

### 4.2. Genes and Variants of CDs in In-House Database

Genes responsible for CDs were searched on OMIM (https://www.omim.org, accessed on 14 December 2021) with “corneal dystrophy” being used as the keyword to search. With reference to the HGMD database and further clarification through the published literature, 15 genes were finally selected (Appendix A). Exome sequencing data for the 15 genes were collected from 10,530 unrelated families with different eye conditions in our cohort, including 37 families with CDs and 10,493 families with other conditions. The exome sequencing data included whole-exome sequencing (WES) or targeted exome sequencing (TES). WES was analyzed using a commercial service. The exome was captured with Agilent SureSelect Human All Exon Enrichment Kit V4 (51189318 bps; Agilent, Santa Clara, CA, USA). Sequencing was performed using an Illumina HiSeq 2000 system (Illumina, San Diego, CA, USA). An average sequencing depth of 125-fold was achieved with this system. Burrows-Wheeler Aligner (BWA; available online at http://bio-bwa.sourceforge.net, accessed on 1 July 2019) was used to map reads against UCSC hg19 (public domain, http://genome.ucsc.edu/, accessed on 1 July 2019). Based on a Bayesian statistical algorithm, SAMTOOLS was used to detect single-nucleotide polymorphisms and indels [41,42]. Variant calling default filters were previously described [43]. TES was based on a next-generation sequencing-based ophthalmic targeted sequencing panel (NGS-based OTSP) to identify causative variants of CDs [44]. To analyze the sequencing data and detect candidate mutants, Strand NGS software (Karnataka, India) was used according to the manufacturer’s instructions [45]. As an alignment reference, the National Center for Biotechnology Information (NCBI) database (NCBI build 19) was used. NCBI RefSeq Gene, NCBI dbSNP 146, dbSNP138 InDels, 1000 Genomes Project, the ExAC database, and Human Genome Mutation Database (HGMD; Professional Version; Qiagen, Germany) were used to annotate and filter single-nucleotide variants (SNVs) and small insertions and deletions (indels) [46].

Using multi-step bioinformatics analysis, pathogenic and likely pathogenic variants of CDs in our cohort were found (Appendix A). First, we excluded low-sequencing-quality variants with coverage less than 5 and common variants with allelic frequency not less than 0.01 in Genome Aggregation Database (gnomAD; http://gnomad.broadinstitute.org/, accessed on 14 December 2021). Then, we predicted the possible impact of missense variants with five online tools, including Rare Exome Variant Ensemble Learner (REVEL; https://sites.google.com/site/revelgenomics/, accessed on 10 July 2022) [47], Combined Annotation Dependent Depletion (CADD; https://cadd.gs.washington.edu/, accessed on 10 July 2022) [48], Sorting Intolerant From Tolerant (SIFT; http://sift.jcvi.org/, accessed on 10 July 2022) [49], Polymorphism Phenotyping v2 (PolyPhen-2; http://genetics.bwh.harvard.edu/pph2/, accessed on 10 July 2022) [50], and Protein Variation Effect Analyzer (PROVEAN; http://provean.jcvi.org/, accessed on 10 July 2022) [51], while the potential splicing effect of the splicing variants and synonymous variants were predicted using the varSEAK database (https://varseak.bio/index.php/, accessed on 10 July 2022), BDGP (https://www.fruitfly.org/, accessed on 10 July 2022), and HSF (https://hsf.genomnis.com/, accessed on 10 July 2022). Furthermore, the allelic frequency of variants was compared with their allelic frequency in gnomAD. Lastly, we further evaluated the selected variants using Sanger sequencing and co-segregation analysis. After that, the selected variants were classified according to American College of Medical Genetics and Genomics and the Association for Molecular Pathology (ACMG/AMP) classification categories [52].

### 4.3. Analysis of Clinical Information

Clinical information was collected from patients with CDs and their participating family members. Diagnosis and classification were based on the IC3D classification [2,3]. All patients had binocular involvement and their recorded clinical information included age at exam, gender, first symptom, visual acuity, result of optometry, anterior segment photograph, etc. Families and patients without available clinical data were excluded from clinical statistics. The clinical information was analyzed, and the genotype–phenotype correlation was summarized. 

### 4.4. Review of Genes and Variants of CDs

The terms “corneal dystrophy” and “gene” were used as the keywords to search on PubMed (https://pubmed.ncbi.nlm.nih.gov, accessed on 2 July 2022), and relevant references in English until June 2022 were included. The responsible genes and variants with CDs as the only or main symptom were collected. The Human Genome Variation Society guidelines (http://www.HGVS.org/varnomen, accessed on 2 December 2022) were used for the variant nomenclature, and the final record of variants was unified into an NM number for readability through the mutalyzer website (https://mutalyzer.nl/, accessed on 2 December 2022). The types of variants were recorded and classified as missense variants, in-frame indels, splicing variants, frameshift variants, nonsense variants, synonymous variants, and non-coding variants. Multiple bioinformatics analysis was implemented to re-evaluate the variants. The possible impact of missense variants was predicted using the five online tools mentioned above: REVEL, CADD, SIFT, PolyPhen-2, and PROVEAN. The possible impact of splicing variants and synonymous variants was predicted with varSEAK online, BDGP and HSF. The frequency of reported variants was compared with their allelic frequency in gnomAD and our in-house exome database. After the contributions of each gene to CDs were clear, the CF threshold was calculated as follows: first, we excluded those enriched in specific populations or subpopulations and the prevalence due to associated genes; then, the CF threshold of the variants in each gene was separately calculated according to its contributions to CDs. The allele frequency in gnomAD of the reported pathogenic variants inherited in an autosomal dominant pattern was compared to their CF threshold. The variants consistent with co-segregation and functional research were also considered. Those variants that did not pass the CF threshold were then subjected to ACMG/AMP classification. The reported families of these genes and variants were roughly divided into three ethnic groups based on the ethnicity of the patients mentioned in the literature. When the patient ethnicity was not mentioned, we used the country of the authors. Phenotypes were recorded and summarized.

### 4.5. Statistical Analysis

For statistical analysis, IBM SPSS Statistic 26.0 was used. Numbers and percentages were used for categorical parameters, while mean, median, and quartile were used to describe continuous parameters. When decimals were involved, two decimal places were used.

## Figures and Tables

**Figure 1 ijms-24-05012-f001:**
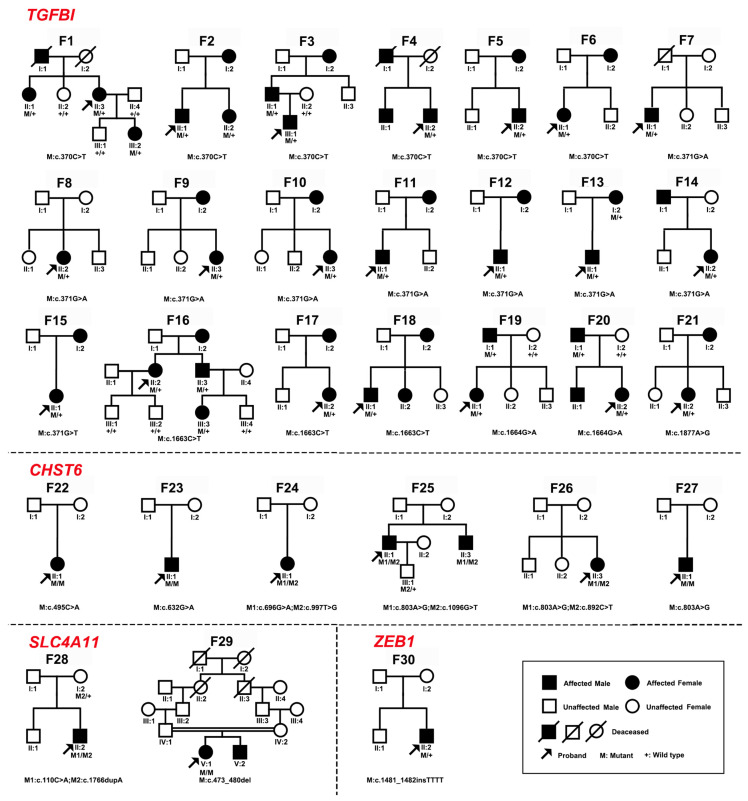
The pedigrees of 30 families with pathogenic or likely pathogenic variants in *TGFBI, CHST6*, *SLC4A11*, or *ZEB1* of CDs in our cohort. The variants of the probands and family members are provided below the pedigrees. M indicates the mutant allele; *+* indicates the wild-type allele.

**Figure 2 ijms-24-05012-f002:**
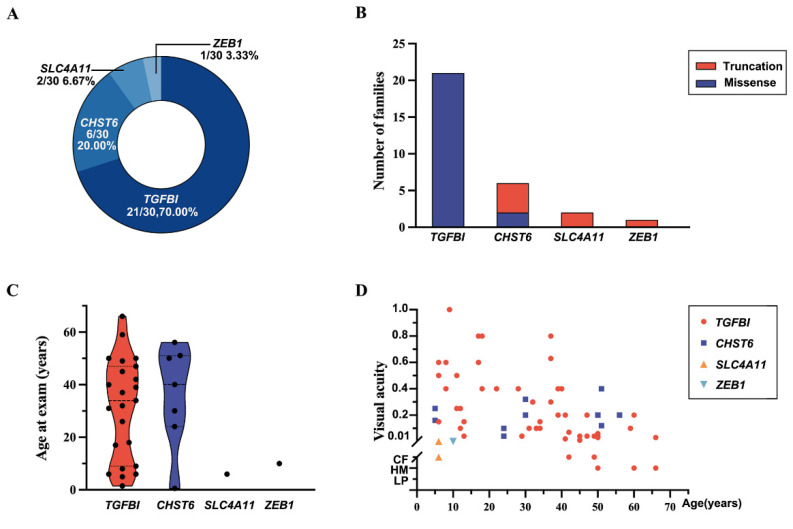
Genetic and clinical features of in-house families of CDs. (**A**) Proportion of in-house families with pathogenic or likely pathogenic variants in *TGFBI*, *CHST6*, *SLC4A11*, or *ZEB1*. (**B**) Distribution of variant types. Truncation variants include splicing variants, frameshift variants, and nonsense variants. (**C**) Correlation of age at exam with variants in *TGFBI*, *CHST6*, *SLC4A11*, and *ZEB1*. (**D**) Correlation of visual acuity with age at exam (including data of both eyes and data at follow-up).

**Figure 3 ijms-24-05012-f003:**
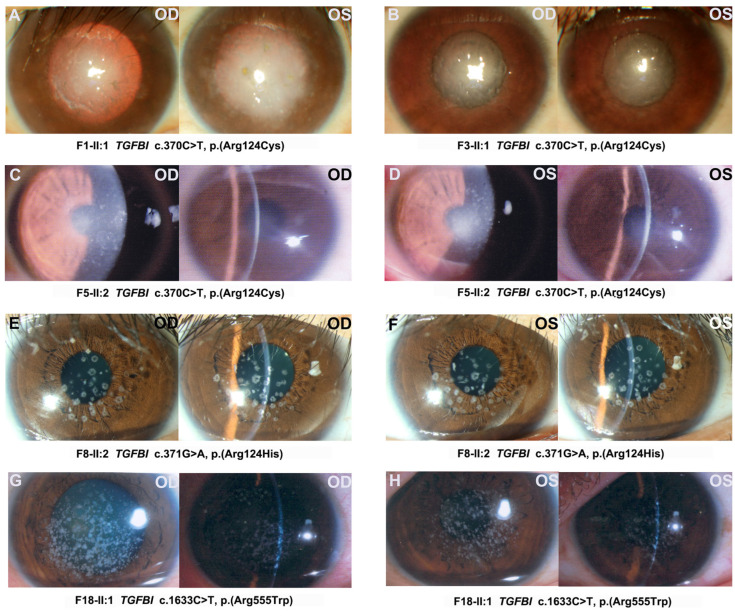
Representative slit-lamp photographs of patients with pathogenic or likely pathogenic variants in *TGFBI*. (**A**–**D**) Patients who showed the phenotype of lattice corneal dystrophy with the same variant, c.370C>T, p.(Arg124Cys). Lattice lines and dot-shaped deposition can be seen in stroma. (**E**,**F**) Patient who showed phenotype of granular corneal dystrophy type 2 with variantc.371G>A, p.(Arg124His). Granular hollow opacities with lattice lines scattered around the stroma. (**G**,**H**) Patient who showed phenotype of granular corneal dystrophy type 1 with variantc.1663C>T, p.(Arg555Trp). Diffuse granular opacities are smaller and denser than in granular corneal dystrophy type 2.

**Figure 4 ijms-24-05012-f004:**
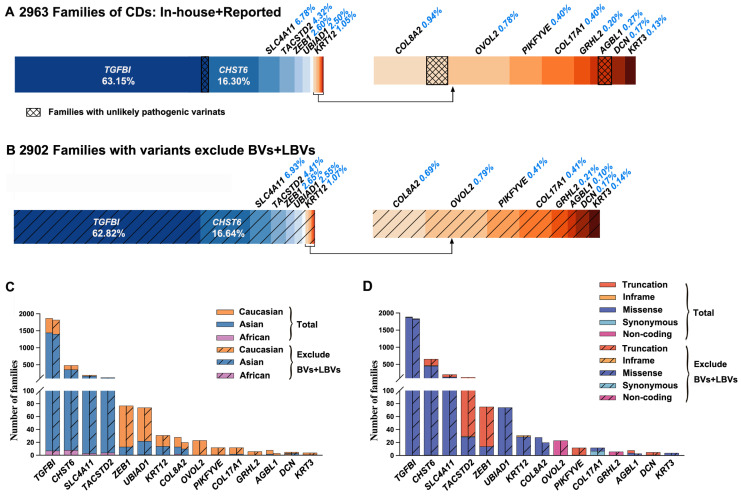
Overview contribution of variants in different genes responsible for CDs and the distribution of variant types as well as ethnic origin of patients. (**A**) Proportion of combined in-house and previously reported families with variants in the 15 genes related to CDs. (**B**) Proportion of families with variants; benign variants (BVs) and likely benign variants (LBVs) in CD-implicated genes were excluded after re-evaluation. (**C**) Distribution of ethnic origin of patients with variants in the 15 genes. The left column indicates total reported variants, while the right column with oblique lines indicates variants with curation after excluding BVs and LBVs. Data included families in current cohort and reported literature reports. (**D**) Distribution of variant types in these genes. The left column indicates total reported variants, while the right column with oblique lines indicates curated variants after excluding BVs and LBVs. Data included families in current cohort and literature reports. Truncation variants include splicing variants, frameshift variants, and nonsense variants.

**Figure 5 ijms-24-05012-f005:**
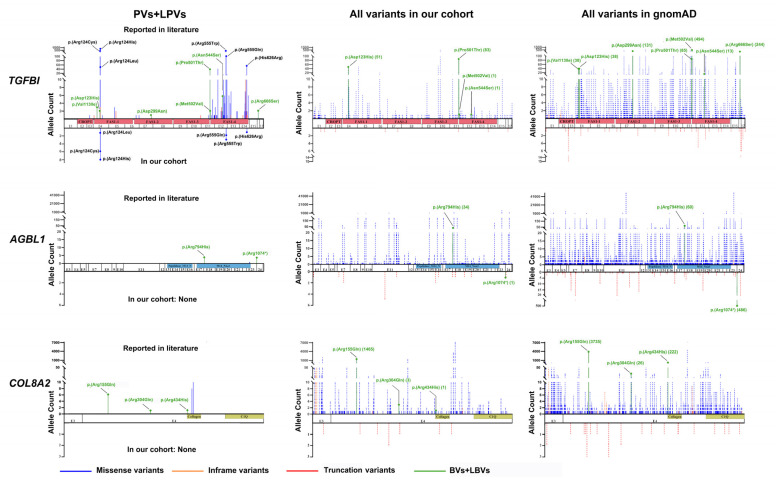
Distribution and frequency of pathogenic variants (PVs) and likely pathogenic variants (LPVs) in *TGFBI*, *AGBL1*, and *COL8A2* in our in-house database and previously reported literature, and variants in our cohort and gnomAD. The green font indicates benign variants (BVs) and likely benign variants (LBVs), and their allele number in our cohort or gnomAD is noted in parentheses. The black font indicates pathogenic or likely pathogenic variants that are hot spots in *TGFBI*. The dotted line indicates variants in population without phenotype of CDs. The solid line indicates variants in patients with CDs. Truncation variants include splicing variants, frameshift variants, and nonsense variants.

**Table 1 ijms-24-05012-t001:** Families with CD phenotype and pathogenic or likely pathogenic variants in *TGFBI*, *CHST6*, *SLC4A11*, *or ZEB1* in current cohort.

Variant	Position	Nucleotide Change	Effect	Number of		Alleles in GnomAD	HGMD	ACMG Rank	ACMG Evidence
				Families	Alleles				
	A. *TGFBI* (Chr:5 NM_000358.2; pLI = 0)								
1	135382095	c.370C>T	p.(Arg124Cys)	6	6	2/280,458	DM	P	PS3+PM1+PM2+PP1+PP4
2	135382096	c.371G>A	p.(Arg124His)	8	8	10/249,054	DM	P	PS3+PM1+PM2+PM5+PP1+PP4
3	135382096	c.371G>T	p.(Arg124Leu)	1	1	/	DM	P	PS4+PM1+PM5+PP1+PP4
4	135392469	c.1663C>T	p.(Arg555Trp)	3	3	/	DM	P	PS3+PS4+PM1+PP1+PP4
5	135392470	c.1664G>A	p.(Arg555Gln)	2	2	/	DM	P	PS3+PS4+PM1+PP1+PP4
6	135396596	c.1877A>G	p.(His626Arg)	1	1	/	DM	P	PS3+PS4+PP1+PP4
	B. *CHST6* (Chr:16 NM_021615.4; pLI = 0)								
1	75513232	c.495C>A	p.(Cys165*)	1	2	/	DM	P	PVS1+PS4+PM4+PP1+PP4
2	75513095	c.632G>A	p.(Arg211Gln)	1	2	1/240,952	DM	LP	PS4+PP1+PP2+PP3+PP4
3	75513031	c.696G>A	p.(Trp232*)	2	2	1/237,152	DM	P	PVS1+PS4+PM4+PP1+PP4
4	75512924	c.803A>G	p.(Tyr268Cys)	23	25	20/275,774	DM	LP	PM3+PP1+PP2+PP3+PP4
5	75512835	c.892C>T	p.(Gln298*)	2	2	4/250,212	DM	P	PVS1+PM2+PM4+PP1+PP4
6	75512730	c.997T>G	p.(Trp333Gly)	2	2	/	DM?	LP	PS4+PP1+PP2+PP3+ PP4
7 *	75512631	c.1096G>T	p.(Glu366*)	1	1	/	/	P	PVS1+PS4+PM4
	C. *SLC4A11* (Chr:20 NM_032034.3; pLI = 0)								
1 *	3218216	c.110C>A	p.(Ser37*)	1	1	4/251,490	/	P	PVS1+PM2+PM4
2	3214820_3214827	c.473_480del	p.(Arg158Glnfs*4)	1	2	/	DM	P	PVS1+PS4+PM4
3 *	3210194	c.1766dupA	p.(Tyr589*)	1	1	/	/	P	PVS1+PS4+PM4
	D. *ZEB1* (Chr:10 NM_030751.5; pLI = 0.97)								
1 *	31809744_31809745	c.1481_1482insTTTT	p.(Lys494Asnfs*12)	1	1	/	/	P	PVS1+PS4+PM4

Notes: * novel variants; chr, chromosome; het, heterozygous; DM, damaging mutation; pLI, probability of being loss-of-function intolerant in gnomAD; P, pathogenic; LP, likely pathogenic.

**Table 2 ijms-24-05012-t002:** Clinical information of patients with pathogenic or likely pathogenic variants in *TGFBI*, *CHST6*, *SLC4A11*, or *ZEB1*.

Patient ID	Genotype	Protein	Gender	Age at Exam	First Symptom	Visual Acuity	Optometry	Phenotype
				(Years)		(OD; OS)	(OD; OS(DS))	
A. *TGFBI* (NM_000358.2)								
F1-II:3	c.[370C>T];[370=]	p.[(Arg124Cys)];[(Arg124=)]	F	49	Poor vision	0.04; CF	NA; NA	LCD
F2-II:1	c.[370C>T];[370=]	p.[(Arg124Cys)];[(Arg124=)]	M	42	Poor vision	CF; 0.07	NA; NA	LCD
F2-II:2	c.[370C>T];[370=]	p.[(Arg124Cys)];[(Arg124=)]	F	31	Poor vision	0.1; 0.1	NA; NA	LCD
F3-III:1	c.[370C>T];[370=]	p.[(Arg124Cys)];[(Arg124=)]	M	9	Ocular pain	1; 1	NA; NA	LCD
F4-II:2	c.[370C>T];[370=]	p.[(Arg124Cys)];[(Arg124=)]	M	50	Poor vision	0.04; CF	NA; NA	LCD
F5-II:2	c.[370C>T];[370=]	p.[(Arg124Cys)];[(Arg124=)]	M	18	Ocular pain	0.4; 0.8	NA; NA	LCD
F6-II:1	c.[370C>T];[370=]	p.[(Arg124Cys)];[(Arg124=)]	F	32	Poor vision	0.3; 0.3	NA; NA	LCD
F7-II:1	c.[371G>A];[371=]	p.[(Arg124His)];[(Arg124=)]	M	59	Poor vision	0.1; 0.1	0.62; 0.75	CD+glaucoma
F8-II:2	c.[371G>A];[371=]	p.[(Arg124His)];[(Arg124=)]	F	29	Poor vision	0.04; 0.04	−7.00; −6.50	GCD2+HMy
F9-II:3	c.[371G>A];[371=]	p.[(Arg124His)];[(Arg124=)]	F	47	Poor vision	0.2; 0.02	−16.00; −10.00	GCD2+HMy
F10-II:3	c.[371G>A];[371=]	p.[(Arg124His)];[(Arg124=)]	F	66	Poor vision	HM; 0.03	NA; −9.00	CD+HMy
F11-II:1	c.[371G>A];[371=]	p.[(Arg124His)];[(Arg124=)]	M	37	Corneal opacities	0.63; 0.63	−1.25; −1.75	CD
F12-II:1	c.[371G>A];[371=]	p.[(Arg124His)];[(Arg124=)]	M	12	None*	0.3; 0.6	−0.25; −0.25	CD+glaucoma
F13-II:2	c.[371G>A];[371=]	p.[(Arg124His)];[(Arg124=)]	M	5	Ocular pain	0.16; 0.25	0.75; 0.75	CD
F14-II:2	c.[371G>A];[371=]	p.[(Arg124His)];[(Arg124=)]	F	50	Poor vision	0.03; 0.06	−7.62; −7.25	CD+HMy
F15-II:1	c.[371G>T];[371=]	p.[(Arg124Leu)];[(Arg124=)]	F	1.5	None*	NA; NA	−1.00; −0.75	CD+strabismus
F16-II:2	c.[1663C>T];[1663=]	p.[(Arg555Trp)];[(Arg555=)]	F	45	Poor vision	0.02; 0.04	NA; NA	GCD1
F17-II:2	c.[1663C>T];[1663=]	p.[(Arg555Trp)];[(Arg555=)]	F	17	Poor vision; ocular pain	0.6; 0.8	NA; NA	GCD1
F18-II:1	c.[1663C>T];[1663=]	p.[(Arg555Trp)];[(Arg555=)]	M	40	Poor vision	0.4; 0.4	NA; NA	GCD1
F19-II:1	c.[1664G>A];[1664=]	p.[(Arg555Gln)];[(Arg555=)]	F	8	Ocular pain	0.4; 0.6	−2.75; −1.00	CD
F20-II:2	c.[1664G>A];[1664=]	p.[(Arg555Gln)];[(Arg555=)]	F	6	Ocular pain	0.6; 0.5	NA; NA	CD
F20-I:1	c.[1664G>A];[1664=]	p.[(Arg555Gln)];[(Arg555=)]	M	39	Ocular pain	0.4; 0.2	NA; NA	CD
F21-II:2	c.[1877A>G];[1877=]	p.[(His626Arg)];[(His626=)]	F	34	Poor vision	0.1; 0.15	−0.25; −0.75	LCD
B. *CHST6* (NM_021615.4)								
F22-II:1	c.[495C>A];[495C>A]	p.[(Cys165*)];[(Cys165*)]	F	44	Poor vision	NA; NA	NA; NA	CD
F23-II:1	c.[632G>A];[632G>A]	p.[(Arg211Gln)];[(Arg211Gln)]	M	30	Poor vision	0.32; 0.2	NA; NA	CD
F24-II:1	c.[696G>A];[997T>G]	p.[(Trp232*)];[(Trp333Gly)]	F	24	Poor vision	0.1; 0.04	NA; NA	CD
F25-II:1	c.[803A>G];[1096G>T]	p.[(Tyr268Cys)];[(Glu366*)]	M	56	Poor vision	0.2; 0.2	NA; NA	MCD
F25-II:3	c.[803A>G];[1096G>T]	p.[(Tyr268Cys)];[(Glu366*)]	M	50	Poor vision	0.2; 0.2	NA; NA	MCD
F26-II:3	c.[803A>G];[892C>T]	p.[(Tyr268Cys)];[(Gln298*)]	F	51	Poor vision; ocular pain	0.12; 0.4	NA; NA	MCD
F27-II:1	c.[803A>G];[803A>G]	p.[(Tyr268Cys)];[(Tyr268Cys)]	M	0.5	Corneal opacities	NA; NA	NA; NA	CD
C. *SLC4A11* (NM_032034.3)								
F28-II:2	c.[110C>A]; [1766dupA]	p.[(Ser37*)];[(Tyr589*)]	M	6	Poor vision	0.06; 0.12	NA; NA	CD
D. *ZEB1* (NM_030751.5)								
F30-II:2	c.[1481_1482insTTTT];[1481=]	p.[(Lys494Asnfs*12)];[(Lys494=)]	M	10	Poor vision	NA; NA	−8.00; −7.50	CD+HMy

Notes: The above data are based on available patients including probands and their family members. M, male; F, female; NA, not available; HM, hand movement; CF, counting fingers; CD, corneal dystrophy; LCD, lattice corneal dystrophy; GCD1, granular corneal dystrophy, type 1; GCD2, granular corneal dystrophy, type 2; MCD, macular corneal dystrophy. None*, Patient presented without symptoms of CDs; F12-II:1, medically consulted for glaucoma; F15-II:1, medically consulted for left eye strabismus; HMy, high myopia.

## Data Availability

The data presented in this study are available upon request from the corresponding author. The data are not publicly available due to ethical privacy.

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
