# Peer review of "Variant Landscape of 15 Genes Involved in Corneal Dystrophies: Report of 30 Families and Comprehensive Analysis of the Literature"

_ijms, 2023, doi:10.3390/ijms24055012_

Round 1

Reviewer 1 Report

In the manuscript Variants landscape of 15 genes for corneal dystrophies: report of 30 families” by  Di Zhu et al. the authors describe spectrum of DNA variants associated with corneal dystrophies from a local cohort of patients and previously reported studies.

My major concern is why the authors introduce “potentially pathogenic variant” (PPV) and “unlikely pathogenic” classification while many diagnostic laboratories and researchers are using “Standards and guidelines for the interpretation of sequence variants: a joint consensus recommendation of the American College of Medical Genetics and Genomics and the Association for Molecular Pathology” by Richards et al (2015) that classify variants as pathogenic, likely pathogenic, variants of unknown significance, likely benign and benign. Some of reported in this study variants represent hot-spot sites (e.g. TGFBI) and are classified as pathogenic and not potentially pathogenic.

Corneal dystrophies (CD) represent a very heterogeneous group in regard to clinical presentation and underlying genetic causes. To-date, genetic causes are not revealed for all subtypes of CD, therefore, describing all CDs together without mentioning which type of cells are affected is not justified. The title is misleading since there are only 4 frequently mutated genes in Chinese cohort.

In general, the manuscript is poorly structured and difficult to follow, several sentences are not finished or repeated. The information in the text and in the Figures is repeated, while clinical data should be separated both in the text and the Figures. Literature data should be presented more clearly, probably in a table form.

(CTG)n expansion in TCF4 gene should be omitted from the manuscript because it is not a single nucleotide change, has technical problems of detection and by the way, present in Fuchs families as well as in sporadic or rather simplex cases.

Chapter on combined data of local cohort and literature data should be omitted or shorten significantly since it does not add any value to the manuscript. Supporting material, such as Figure 5 should be modified.

Reviewer 2 Report

Title: Variants landscape of 15 genes for corneal dystrophies: report of 30 families

1. The American College of Medical Genetics and Genomics 2022 (ACMG)-recommended five variant classification categories (pathogenic, likely pathogenic, uncertain significance, likely benign, and benign) have been widely used in medical genetics. In this study, what is the definition of potential pathogenic variant (PPV)? It seems to be an old term or a term used arbitrarily, so the author's consideration is needed.

2. How did you match NM number in comparative studies in the literature outside your cohort?

3. Method 4.1 The reviewer think it is an unnecessary paper self-citation (in our previous study[32]).

4. Method 4.2 the author should provide methods in detail. WES (x30)? NGS lacks fundamental details regarding which machine data was generated. For better understanding, Bioinformatics procedures summary as figure.

5. It should be indicated how many people in the cohort the data were analyzed for this study.

6. It is necessary to improve the completeness of Figure 5. At the current stage, it is difficult to distinguish the green font without even seeing the letters.

7. For rare disease sequencing data, it is necessary to be cautious about disclosing anonymized IDs (patients id) in Table 2. In addition, providing of corneal pachymetry or endothelial cell counts is more informative for clinical features.

Round 2

Reviewer 1 Report

Review

“Variants landscape of 15 genes involved in corneal dystrophies: report of 30 families and comprehensive analysis of the literature” by Zhu et al.

Extensive revision was done by the authors however some issues are not resolved.

1.     “Clean” version was not provided which made review process difficult, especially without pages and lines numbering

2.     “predominated” -  consider synonym

3.     check using the same phrase too often e.g. “in the era of next generation sequencing” or “genomic medicine” twice on page 2

4.     page 3 – the same tense – demonstrate/revealed

5.     missense variant is defined by DNA,  truncation is defined by protein, better to say nonsense and frameshift

6.     analysis was conducted

7.     Table 1 – good!

8.     The first symptom in 68.75% (22/32) patients was poor vision

9.     % - use even numbers without decimals

10.  Coexist – is it correct? Co-exist instead

11.  Instead for “For TGFBI, patients harbored variant c.370C>T/p.R124C presented LCD (Lattice Corneal Dystrophy), patients harbored variant c.371G>A/p.R124H presented with GCD2 (Granular corneal dystrophy type 2), and patients harboringed variant c.1663C>T/p.R555W showed the phenotype of GCD1 (Granular corneal dystrophy type 1)” consider

“The patients with TGFBI c.370C>T/p.R124C presented LCD (Lattice Corneal Dystrophy) phenotype, the patients, carriers of TGFBI c.371G>A/p.R124H had the GCD2 (Granular Corneal Dystrophy, type 2), and the patients with TGFBI c.1663C>T/p.R555W showed the phenotype of GCD1 (Granular Corneal Dystrophy, type 1)”

12.  Check HGVS sequence variant nomenclature  and use it accordingly, “comma” should be used instead of “/” c.1663C>T, p.(R555W). Three- letters aminoacis code is preferable

13.  Page 7 - the proband of five families  - is it proband of five families or in five families? What are these families?

14.  The numbers are confusing in the sentence “Furthermore, the proband of five families had both high myopia and corneal dystrophy. Among eight 8 probands with the c.371G>A/p.R124H variant in TGFBI, four (4/8, 50.00%) had both granular corneal dystrophy and high myopia, and two (2/8, 25.00%) had both granular corneal dystrophy and glaucoma. Omit the sentence “Furthermore, the proband of five families had both high myopia and corneal dystrophy.”

15.  Should be revised – “The genes causative for CDs in Asians were mainly TGFBI, CHST6, SLC4A11, and TACSTD2, while those in Caucasian were mainly TGFBI, CHST6, TACSTD2, and ZEB1.” At least three genes are same.

Reviewer 2 Report

After revision, the manuscript minimized of the concerns of reviewer. It might be a good paper for following researchers. Good luck.

Author Response

Thanks.

Round 3

Reviewer 1 Report

I am satisfied with the response provided by the corresponding author.